# Hyaluronan-Based Nanofibers: Fabrication, Characterization and Application

**DOI:** 10.3390/polym11122036

**Published:** 2019-12-09

**Authors:** Petr Snetkov, Svetlana Morozkina, Mayya Uspenskaya, Roman Olekhnovich

**Affiliations:** Institute BioEngineering, ITMO University, Kronverkskiy Prospekt, 49, St. Petersburg 197101, Russia; Morozkina.Svetlana@gmail.com (S.M.); mv_uspenskaya@mail.ru (M.U.); r.o.olekhnovich@mail.ru (R.O.)

**Keywords:** biocompatibility, biodegradability, cell growth, drug delivery, electrospinning, hyaluronan, nanofibers, nanoparticles, polymer solution, spinnability

## Abstract

Nano- and microfibers based on biopolymers are some of the most attractive issues of biotechnology due to their unique properties and effectiveness. Hyaluronan is well-known as a biodegradable, naturally-occurring polymer, which has great potential for being utilized in a fibrous form. The obtaining of fibers from hyaluronan presents a major challenge because of the hydrophilic character of the polymer and the high viscosity level of its solutions. Electrospinning, as the advanced and effective method of the fiber generation, is difficult. The nano- and microfibers from hyaluronan may be obtained by utilizing special techniques, including binary/ternary solvent systems and several polymers described as modifying (or carrying), such as polyethylene oxide (PEO) and polyvinyl alcohol (PVA). This paper reviews various methods for the synthesis of hyaluronan-based fibers, and also collects brief information on the properties and biological activity of hyaluronan and fibrous materials based on it.

## 1. Introduction

A unique biopolymer, hyaluronan (hyaluronic acid, HA) is a naturally occurring biopolymer used in medical applications ranging from cataract surgery and hydrophilic coatings for post-surgical adhesion prevention to cancer treatment [1].

Hyaluronan has recently gained more attention for wound dressing formation due to its natural presence in the extracellular matrix (ECM) and involvement in the inflammation and proliferation stages of wound healing. HA is currently clinically used in wound dressings [2], skin substitutes [3], and joint lubricants [4]. Hyaluronan-based biomaterial (HYAFF-11), a commercialized biocompatible HA-matrix, has improved mechanical integrity upon swelling by implementing an esterification crosslinking process [2].

Nanometer sized materials have become increasingly popular in recent years in the field of tissue engineering due to their ability to be fabricated into scaffold materials and to be designed as drug and protein delivery systems [5].

Nanofiber scaffold formulations (diameter less than 1000 nm) were successfully used to deliver the drug/cell/gene into the body organs through different routes for an effective treatment of various diseases. The unique characteristics of nanofibers, such as higher loading efficiency, superior mechanical performance (stiffness and tensile strength), controlled release behavior, and excellent chemical stability, help in the delivery of plasmid DNA, large protein drugs, genetic materials, and autologous stem-cells to target sites [6].

One of the most effective methods for generation of fibers with diameters from 10 nm to several hundred nanometers is electrospinning. By utilizing such a technique, three dimensional scaffolds could be obtained from natural polymers, which are ideal tools due to their biocompatibility and biodegradable properties [7]. Such scaffolds can be used for tissue engineering to create or repair damaged tissue or organs by the fabrication of a three dimensional scaffold that combines biological agents and biomaterials that can support cell proliferation [8].

Due to a high versatility of electrospinning method, hundreds of polymers, especially if they are soluble in high-volatile organic solutions, have been easily electrospun to produce nanofibers for various applications. At the same time, many polymers have not been successfully electrospun. Highly concentrated polymer solutions are very difficult to electrospin, as viscosity is one of essential parameters in electrospinning [9].

By contrast, HA-solutions are not highly-concentrated; however, have a high viscosity level. Thus, an HA solution with content 10 mg/mL has a viscosity which is 5000 times that of water [10]. Hyaluronan has been reported to form a secondary structure which is reminiscent of flat belts coiled, which is explained by intermolecular hydrogen bonds. HA may form coils, which are considered tertiary structure [11,12,13,14,15]. In the case of increased concentrations (of 1 mg/mL and above) polymer helixes and coils could overlap each other, resulting in the formation of third-dimensional interpenetrating polymer networks and increasing solution viscosity [16].

Nanofibers in the range of 10 to 2000 nm diameter can be achieved by choosing the appropriate polymer solvent system [17]. Typical solvents for the electrospinning of biopolymers are formic acid (FA), dimethyl formamide (DMF), trifluoroacetic acid/dimethyl chloride (TFA/DMC), water, DMF/toluene, dimethyl acetamide (DMA), sulfuric acid (SA), phenol, etc.

Hyaluronan is widely used in biomedical applications; however, the data for HA nanofibers are very limited. The high viscosity levels, low evaporability, and high surface tension of HA-solutions make it challenging to electrospin, as those are important parameters for successful nanofiber production.

In the first report about HA fiber creation [18], the group successfully produced HA nanofibers with electro-blowing assisted procedures. After the testing of several solutions through numerous trials, the researchers concluded the following parameters as optimal conditions for the process: solution concentration: 1.3–1.5 *w*/*v*%; solution viscosity: 3–30 Pas; flow rate: 20 to 60 µL/min. Consequently, these parameters were deemed inadequate as they were not able to be replicated over a series of experimental trials. To overcome this issue, the electro-blowing assistance was introduced, and from further testing, the researchers found that at a blowing rate of 57 °C at a 70 ft^3^/h flow rate, the nanofiber production was of high quality and consistency.

In their second publication [19], two methods to fabricate water-resistant HA nanofibrous membranes without the use of reactive chemical agents were demonstrated with the exposure of HA membranes in HCl vapor, followed by a freezing treatment at −20°C for 20–40 days; and the immersion of HA membranes in an acidic mixture of ethanol/HCl/H_2_O at 4°C for 1–2 days. Although both methods could produce hydrophilic, substantially water-resistant HA nanofibrous membranes (the treated membranes could keep their shape intact in neutral water at 25 °C for about 1 week), the immersion method was shown to be more versatile and effective.

It is expected that HA-based fibrous materials will become widely used, similarly to, for instance, highly-effective filters and filtration nanofibrous systems which have been a growing field for over a quarter of a century [20].

Regardless, the interest of the research community in modern micro- and nanofibers from different biopolymers is known, and very few reviews have been concerned with said issue. Our paper is one of the first to collect data of the most attractive hyaluronan-based fibers—its methods of fiber formation, properties, and applications.

## 2. Key Techniques for Polymer Fiber Fabrication

Polymer fibers can be obtained several ways. Some of them, such as wet spinning and electrospinning, which are considered in detail below, became a frequent practice and are widely utilized for fabrication of various materials, from air and water filters to scaffolds and wound dressings. The above-mentioned methods are key approaches for the investigation and fabrication of fibers based on hyaluronan. They allow one to provide the production of polymer fibers with different required and mutable parameters: sponginess, geometry (predominantly, diameters), regularity, etc.

The drawing technique [21] and dry spinning method [22] are well known; however, are not widely used for polysaccharide fibers, including hyaluronic acid-based ones. The first one has low productiveness, and it is employed for research [23]. The second one is difficult to realize because of the water-soluble nature of hyaluronan, and, probably could be utilized for hydrophobic HA-derivatives, which are soluble in volatilizable organic solvents.

There is one interesting technique for the preparation of HA-based fibers [24]. Firstly, the polymer films prepared from an HA solution were dried on a clean glass surface. After that, the film obtained was cut into streaks approximately 1–2 mm width. Later cords of individual streaks were stretched under constant load at a relative humidity of 75%–85%.

### 2.1. Wet Spinning Method

Hyaluronan, as a member of polysaccharide family (actually a hetero-polysaccharide), cannot be fluxed; consequently, fibers cannot be fabricated by the melt spinning process, as opposed to, for instance, polyester and polyethylene terephthalate [25,26]. By contrast, the wet spinning method adopted from the textile industry allows researchers to obtain biopolymer fibers with a set diameter. The process of wet spinning is carried out as follows: the polymer is dissolved into an appropriate water-based or organic-based solvent medium and then solution is extruded through a spinneret (jet, needle, set of needles, nanonozzles, etc.) into a coagulating bath (or subsequently positioned baths) in which fiber formation is being carried out [27]. In the wet spinning process, the jet or spinneret is immersed into a liquid, which either diffuses throughout the solvent or reacts with the stretchable fiber [28]. The scheme of the wet spinning process may include washing bath, drawing bath, drying section, and winding machine [29].

Template synthesis [30] relates to the wet spinning process and includes a spinneret (membrane) with nanosized holes.

The principal scheme of wet spinning process is demonstrated in Figure 1.

Obviously, the diameters of fibers obtained directly depend on jet diameter, and the stretching stage renders it possible to decrease diameters and enhance the mechanical properties [31] of fibers.

The so-called dry-wet spinning technique is a method in which the moving stream of a polymer solution is flown through the air, spun with the range from a several millimeters to several centimeters before the coagulation bath [32].

The technique of wet spinning from hyaluronan is well known. For instance, Rupprech A. [33] used a 2.5–3.0 mg/mL solution of potassium hyaluronate in 0.1 M KCl which was extruded with a feed rate from 9 to 30 mL/h through a spinneret with 720 cylindrical channels, each 70.0 μm in diameter, into a precipitation bath containing 70%–80% ethanol with 0.1 M KCl. During coagulation, the polymer fibers were stretched and collected into the rotating Teflon-coated cylinder. After the collecting, the individual and oriented fibers were agglutinated into a polymer film. The authors did not measure diameters and tensile strengths of the fibers obtained; for this reason the results could not be compared with other researches’ data.

As mentioned above, hyaluronan possesses high levels of hygroscopicity and water solubility. Thus, the fibers obtained have a low resistance against aqueous solutions and susceptibility to high relative humidity. Nonetheless, some researchers synthesized a variety of HA-derivatives, which have low solubility in water.

For example, Pitucha T. et al. [34] utilized hyaluronan acylated by fatty acids, such as palmitic, oleic, and stearic acids, with different substitution degrees. The soluble hyaluronan was utilized for the obtaining of core-shell fibers. As a result, the fibers with the tensile strength from 0.05 to 0.1 N/tex and breaking elongation 8.1% ÷ 15.4% were successfully obtained. The authors highlighted that physicomechanical properties of the fibers based on HA directly depend on the relative humidity of the air: the fibers started to be crack-sensitive and broke during formation or testing when the humidity is low. Moreover, the iron oxide nanoparticles and octenidine dihydrochloride were used as functional agents and medical drugs (against Gram-positive and Gram-negative bacteria), respectively. The successful addition of medical agents into fibers is the way to a new drug delivery system and personal healthcare. The fibers obtained had a high swelling rate: the fiber from HA-derivatives could have three-fold increase in diameter over a few minutes. By contrast, the native HA-core begins to dissolve immediately after immersion.

Vojtech Zapotocky et al. [35] utilized palmitoyl-derivative of hyaluronic acid. Modified HA was dissolved in 50% isopropanol with further transfer into the coagulation bath containing a binary solution of lactic acid and isopropanol (volume ratio 20:80). The resultant fibers were washed with alcohol, stabilized in acetone, and dried at ambient conditions. After synthesis, the fibers were subsequently processed into a warp knitted fabric. Tensile strength of the fibers obtained was from 0.057 to 0.093 N/tex, breaking elongation—from 9.7% to 18.9%. The authors also demonstrated that the fibers had low degree of endotoxicity and general bioburden, which allows one to modify the palmitoyl-hyaluronan fibrous materials by fibronectin/fibrinogen, resulting in improved adhesion and proliferation of normal human fibroblasts in vitro.

Another method for the fabrication of water-resistant fibres with HA is to create a polyelectrolyte complex between negatively charged hyaluronic acid and a positively charged polymer. For example, Jou [36] presented multilayer fiber based on polyethylene terephthalate. Polyethylene terephthalate fibers were treated with ^60^Co-γ-ray and grafted with acrylic acid, and then were grafted with chitosan via esterification. At last, hyaluronic acid was immobilized onto chitosan-grafting fibers. It is interesting that a polyethylene terephthalate core provides high stress–strain properties to the fibers. The authors did not measure the morphology and physicomechanical properties of fibers, but emphasized surface density of grafted functional groups, activated partial thromboplastin time (APTT), antibacterial activity, and cell proliferation. The fibers demonstrated an antibacterial effectiveness and improved the cell proliferation of fibroblasts.

The next studies [37,38,39] describe a three-step method of fiber creation. The chitosan solution was spun through the stainless steel spinneret into the first coagulation bath with CaCl_2_ dissolved in 50% aqueous methanol solution; then, formed fibers were flown through the second bath with a 50% aqueous methanol solution, and, finally, through the third bath containing hyaluronan dissolved in 50% aqueous methanol solution. After creation, the fibers were stretched, drayed, and winded. Then the fibers were intertwined into a scaffold to be investigated. The authors put emphasis on the biological analysis: cell adhesion, proliferation, and morphological, and immunohistochemical analyses. These studies demonstrated that the chitosan-based/HA fibrous material had significantly better physicomechanical properties and adhesion to fibroblasts and extracellular matrix (ECM) products. The results indicate that hyaluronan could enhance the tensile strength of the fibers and have the beneficial effect of adhering fibroblasts in a scaffold.

The property of hyaluronan to form a polyelectrolyte complex with chitosan has applications in fiber technology and also in the development of hydrogels [40], colloid systems [41], nanoparticles [42,43], etc. Thus, such polymer complexes have broad fields of application and open the way to effective regenerative and targeted medicine.

Hyaluronan fibers spun by wet technology are known and patented. For example, the method offered by Burgert Ladislav et al. [44] includes direct extrusion of HA-solution into a coagulation bath containing alcohol, acid, and water. The authors considered that the wet-dry spinning process is also possible and the air gap between the jet and the bath could be varied from 1 to 200 mm.

HA-based fibers with specific additives are also known. For instance, Ting Zheng et al. [45] developed hyaluronan-based fibers filled by multiwalled carbon nanotubes. Once again, hyaluronan was dissolved in distilled water with a consequent addition of multiwalled carbon nanotubes. Later, spinning solution was extruded into a precipitating liquid contained CaCl_2_ dissolved in 70% ethanol. The obtained fibers were removed from the bath and washed with alcohol and water. The fibers were dried in ambient conditions under tension. It is worth noting that the authors were the first to analyze the functional relationship between the injection speeds (as a set parameter) and morphologies and tensile strengths of the fibers obtained. The authors also investigated the flexibility—one of the most important technological and operational parameters.

Considering that the authors utilized different methods of analysis, the comparison between results obtained is hampered. To obtain repeatable and comparable results it is essential to fabricate the fibrous materials using the above-mentioned ways and to carry out an analysis of physicomechanical, physicochemical, biological, toxicological, and morphological properties using equal test conditions.

All things considered, it seems reasonable to assume that wet spinning is a simple method to obtain individual fibers or a bundle of fibers from hyaluronan. Moreover, it is expected that wet spinning renders it possible to scale the process from the small laboratory unit to an enormous industrial installation, similar to textile manufacturing. Nonetheless, the wet spinning method has limitations. Firstly, the usage of volatile alcohols (methanol, ethanol, isopropanol, etc.) needs special regenerative systems and personnel protection, which drastically raises the price of the process line. Secondly, the final fibrous material could not be obtained directly by wet spinning, as it needs the textile machine, which complicates the process.

### 2.2. Electrospinning

Electrospinning (electroforming, electrohydrodynamic jetting) is a cross functional, challenging, and advanced technology for the fabrication of polymer nano- and microfibers and materials based on such fibers being used for versatile industrial and scientific sectors, and their biomedical applications are of special interest. The application areas of electrospun fibrous materials are as follows: wound dressing, ambustial healing, drug delivery systems, tissue regeneration and engineering (dermal, bone, neural, cardiac, vascular, etc.), nanomedicine, target cancer treatment, diagnostic systems, biosensors, etc. [46].

Note, that electrospinning in comparison to wet spinning allows one to obtain the finished fibrous materials (canvas of fibers), which could be utilized and fabricated. In other words, the individual fibers or fiber bunch obtained by the wet spinning method needs to be reprocessed, and in the case of the scaffold production, be intertwined into fibrous, nonwoven fabric.

Electrospinning is a process related to the exposure of electrostatic forces to the polymer solution. Electrospinning is close to electrospraying, which is an effective and simple method of obtaining nano- and microparticles from polymer solutions atomized via electrostatic forces. Both electrospinning and electrospraying also could be named electrohydrodynamic jetting [47]. The method of liquid dispersing and equipment designed for this was separately patented in 1902 by William James Morton [48] and John F. Cooley [49] respectively. Polymer fibers obtained via electrospinning process had a little worldwide scientific and industrial interest until the 1990s; after that the researchers recognized the significant potential of the electrospun fibers [17,50]. Nevertheless, first electrospun non-woven fibers as wound dressing material were patented by Anton Formhals in 1934 [51,52]. Air Petryanov filters developed in the USSR in 1939 are an example of the first effective commercial application of an electrospun product with industrial production [53,54].

The schematic representation of the electrospinning/electrospraying process is demonstrated in Figure 2.

Obviously, a high current voltage source is required. The electric voltage value directly depends on equipment and at average is equal to the interval from 1.0 to 50 kV or higher. The second key element of the spinning process is the delivery system with a metal needle, spinneret, or nozzle at the end. The third main component is a metal collecting electrode, which could have various forms: flat plate, drum, mandrel, disk, etc. In the electrospinning process a high voltage is utilized to obtain an electrically-charged stream of polymer solution escaping from the nozzle. The first electrode is connected to the bin with the spinning solution (or to the needle at the end of the delivery system) and the second electrode is conjugated with the collecting screen, and usually the collector is grounded [55]. Before fibers reach the collecting electrode, the solvent evaporates from them, and later fibers are collected as a web of fibers [56]. Importantly, is that a variety of technological parameters of electrospinning process could change the degree of fiber attachment.

The presence of the high voltage source requires stable energy power consumption, specific equipment, and extraordinary safety measures. The working chamber must be closed during the electrospinning process and has to be blown through with air. Moreover, the exhaust air must be filtrated or regenerated to avoid the ambient air pollution by vapors of organic solvents. For these reasons electrospinning could not be easily upscaled from a laboratory to industrial unit. Nonetheless, we found several manufacturing applications which allow one to obtain the fibrous materials with width from 1000 to 2000 mm at quantity of approximately 5000 m^2^ per day.

As mentioned before, hyaluronan is a highly-water soluble natural polymer. Due to the high level of electrical conductivity of the polymer solution and a low level of evaporability, the performance of the electrospinning process is difficult because of the possibility of the “electric breakdown”; in other words, circuit faults between the needle and collector. Wet (not dried, with residual solvent) fibers could act as electrical conductors between the electrodes, resulting in short-circuiting. Smart and modern electrospinning systems could stop the process, if any wrong behavior was detected.

Hyaluronan, like other biopolymers, is not soluble in volatile organic solvents, which is strongly encouraged for the electrospinning process. Nonetheless, some HA-derivatives could be dissolved in organic solvents, but any impacts on hyaluronan are able to cause the depolymerization [16].

Another key thing to remember is a high viscosity level of hyaluronan solutions, which was discussed in the introduction. There are only several studies which describe the hyaluronic-acid based fibers’ preparation attempts.

Thus, Junxing Li et al. [57] demonstrated that the utilization of aqueous hyaluronan solutions results in obtaining irregular beads with few fibers. However, obtaining fibers strongly depends on the distance between the needle and the collecting electrode. Moreover, the electrospinning process is not stable and satisfactory. Further, the authors using dimethyl formamide (DMF) and distilled water (DW) considered that the proper volume ratio of DMF to DW is from 1.5 to 0.5. They added a gelatin solution with polymer concentrations from 1.5 to 7.5 *w*/*v*% into a hyaluronan solution and demonstrated that gelatin acts as a surfactant, which decreases the surface tension of the HA-solution, and consequently, improves electrospinning processability. The fibers with the average diameter from 120 nm to 500 nm were successfully obtained.

The next study [58] confirms the availability of fibers from aqueous solutions. The authors utilized surfactant—cocamidopropyl betaine, which reduces the surface tension of the solution. The authors analyzed the relationship between technological parameters (flow rate, spin length, and voltage) and the average diameter of nanofibers, which was from 58 nm to 1016 nm. Note that this study also demonstrated practical results of pigs’ wound healing.

Yang Liu et al. [59] utilized a three-component solvent containing distilled water, formic acid, and DMF with a weight ratio 25:50:25. Even a small additive of organic or mineral acid starts the process of the molecular-chain scission (depolymerization) [16]. However, the authors successfully obtained nanofibers with diameters ranging from 30 to 50 nm. The research investigated various solutions, including water based, and only one resulted in positive spinnability.

Yao Shenglian et al. [60], apart from solvents properties, highlighted an influence of ambient relative humidity on the electrospinnability of the hyaluronan solutions. A critical value of humidity allowing them to successfully obtain uniform hyaluronan fibers with similar diameters and morphologies was equal to 8%. When the relative humidity is increased to 18% and above, the fibers underwent fusion. Thus, the authors obtained the fibers with average diameters from 28 nm to 132 nm. The authors highlighted that the fibers could be stabilized by hydrochloric acid solution.

Eric Brenner et al. [61] utilized DMF/distilled water alkali solutions. The authors reported that after preparing the hyaluronan solution in the 4:1 mixture of NaOH:DMF, the electrospinning became hindered after 30 min, which could be explained by degradation of hyaluronic acid in the highly basic media. This fact is confirmed by Atoosa Maleki and coauthors [62]. Pure hyaluronan fibrous material with an average diameter from 27 to 352 nm was obtained.

Nevertheless, the possibility for the obtaining of fibers based on native hyaluronan from water/dimethyl sulfoxide solutions without additional polymers like PEO and PVA was firstly demonstrated in our primary study [63]. The HA-fibers with the average diameter of 309 nm were obtained.

The majority of researchers noticed that hyaluronan solutions have a low evaporability and a very high viscosity level and surface tension, which significantly hinder the process of electrospinning. Moreover, it is emphasized that the narrow hyaluronan concentration window is equal to 1.3–1.5 *w*/*v*% in which the electrospinning process may be realized [18]. The authors modified the electrospinning technique by air-blowing assistance, termed “electroblowing,” expanding the capability of the electrospinning and overcoming the shortcomings of the high viscosity of hyaluronan solutions. Furthermore, the researchers added ethanol, being a liquid with a lower boiling point, higher vapor pressure, and lower surface tension (than water), and found that the fiber content could be increased by introducing alcohol. At the same time the spinning jet remained unstable and the improvement was not serious. The authors demonstrated the strong dependence between the air temperature and stability of electro-blowing process: the stable fibers with the average diameters equal to 49 and 74 nm, respectively, could be fabricated only at 47 °C and 57 °C. However, in the first case a small fraction of beads was obtained apart from the smooth fibers. The study was extended in the next publication [19].

The next method helping to fabricate hyaluronan-based fibers is the utilizing of water-soluble modifying polymers named carrying polymers. The most usable polymer is polyethylene oxide (PEO) [64,65]. In the first case, 50 wt% aqueous acetic acid was utilized as a solvent that could result in hyaluronan depolymerization. However, PEO/HA core-shell nanofibers with the diameter from 90 to 460 nm were obtained by the electric field induced phase separation of polymers during the electrospinning. The second study describes PEO nanofibers with diameters from 63 to 159 nm containing hyaluronan and cationic aminoglycosidic antibiotic kanamycin. Note that such PEO based fibers with a small amount of HA are far easier to obtain in comparison with pure HA-based fibers.

The next scheme involves electrospinning with postprocessing. One of the most usable subsequent exposures of hyaluronan (not only for fibers) is crosslinking. For example, the authors of [66] describe hybrid porous cross-linked fibers based on HA/collagen obtained from NaOH/DMF solutions. The electrospinning technique was combined with a sieved NaCl salt particulates dropping process with subsequent cross-linking by 1-ethyl-3-(3-dimethylaminopropyl) carbodiimide hydrochloride (EDC) dissolved in acetone/H_2_O (95/5) and salt leaching. Initial fibers (before crosslinking) had mean diameters from 203 to 394 nm. The deposition of salt as a porogen during electrospinning results in scaffolds with a macroporous and nanofibrous geometry, which improves the cell adhesion and proliferation.

The study [67] describes the same method of the hyaluronan fiber creation with the average diameters 100–300 nm from NaOH/DMF solutions, without salt dispersion. Cross-linking was performed after the fibers’ preparation. The authors also gave a reason for utilizing EDC as an applicable cross-linker: EDC is not included into the macromolecules [68], but is able to modify the side functional groups of biopolymers, resulting the bond formation between –OH and –COOH groups [69]. The authors also successfully conjugated Au nanoparticles with the fibers and determined cellularity and cell attachment enhancement.

It is known that the dual-syringe, reactive electrospinning technique results in the obtaining of cross-linked hyaluronan fibers [70,71]. Instead of native hyaluronan, the low-molecular-weight derivative, 3,3′-dithiobis(propanoic dihydrazide)-modified hyaluronan, was used. In comparison with the above-mentioned studies, controllable crosslinking occurred during the electrospinning process using the dual-syringe technique. Then, PEO was used as the carrying polymer and a viscosity modifier, and was selectively removed with water after electrospinning. Initial hybrid fibers had diameters from 75 to 105 nm, but after removing PEO and lyophilizing the matrix, the mean diameter of the fibers shifted to 82–138 nm.

The novel research in [72] describes nanofibers with diameters from 60 to 360 nm based on HA/cyclodextrin. In this case, polyvinyl alcohol (PVA) was added as a carrier polymer, and 2-hydroxypropyl-beta-cyclodextrin was utilized as a stabilizer of electrospinning process. A noteworthy detail is that the authors added crosslinker reagents (EDC and N-hydroxysuccinimide (NHS)) in the spinning solutions 30 min before the electrospinning, and the activation of the crosslinking reaction in the nanofibers obtained was promoted in an oven. Moreover, the authors utilized Naproxen, which was loaded within fibrous material either by solution impregnation or by supercritical carbon dioxide assisted impregnation.

Apart from single-layer fibers and scaffolds based on them, the bilayer scaffolds were successfully developed and investigated. To obtain fibers from hyaluronan, Amit Chanda et al. [73] also utilized PEO/HA aqueous solutions with excess of PEO compared to HA equal to 25 times. HA/PEO fibers with average diameters from 75 to 197 nm were obtained. At the same time, chitosan/polycaprolactone dissolved in formic acid/acetone mixture were used as the second layer of the scaffold. Moreover, two layers were exposed to glutaraldehyde vapor for subsequent interlayer crosslinking reaction.

Chitosan could be utilized as a component of a polyelectrolyte complex with hyaluronan. Fibers based on such a complex have been obtained [74]. The strongest electrostatic interaction between amino cationic groups of chitosan and the anionic carboxyl group of hyaluronic acid is the reason for polyelectrolyte complex creation [40]. Moreover, if a complex consists of equal amounts of oppositely charged polyions, it has zero overall net charge and could be precipitated from the solution [75]. Nevertheless, the authors successfully obtained nanofibers with diameters from 80 to 330 nm from mixture of HA aqueous/formic acid (25/75 *w*/*w*) solution and chitosan aqueous/formic acid (20/80 *w*/*w*).

A recent study [76] described that the bilayered, nanofibrous material consisted of a chitosan/PEO layer with average diameters from 360 to 420 nm and an HA/PEO layer with average diameters from 370 to 650 nm. Step-by-step electrospinning allowed them to form the polyelectrolyte complex, which could stabilize the material and decrease the solubility without a crosslinking technique. The authors demonstrated that the bilayered, chitosan-hyaluronan fibrous material had better biocompatibility than the chitosan matrix.

The independent approaches to obtain electrospun fibers based on hyaluronan are the creations of various derivatives. For example, in the article [77], a series of partially hydrophobic and water soluble alkyl ether derivatives of hyaluronan were synthesized; however, they did not form by electrospinning.

The study [78] describes electrospinnability of 3,3′-dithiobis(propanoic dihydrazide)-modified hyaluronan in the presence of PEO and cross-linker—poly(ethylene glycol) diacrylate (PEGDA). Cross-linking was carried on after the electrospinning using an Eppendorf micropipette with a 25% amount of the electrospinning solution. The fibers with diameters from 50 to 70 nm were obtained.

Another hyaluronic acid derivative is known [79]. The solutions with norbornene-functionalized hyaluronic acid, albumin, PEO, UV-initiator, and dithiothreitol were electrospun onto thiolated coverslips. PEO was utilized as a carrier polymer; albumin was included for limitation of non-specific absorption of thiolated peptides in the patterning process. A crosslinking process was performed under UV exposure. The initial dry fibers had initial diameters 220 ± 50 nm, and after swelling the fibers size increased to 740 ± 140 nm.

Hyaluronan is also used as additional hydrogel-forming polymer; for example, poly(L-lactic acid) electrospun nanofibers could be combined with HA-hydrogel by injecting of the last into the centers of nanofiber coils [80].

There are bilayered fibrous materials based on poly(L-lactic acid) as the first layer and hyaluronan as the second layer [81], and the layers were fabricated one after another, likely to chitosan/polycaprolactone-HA bilayer scaffolds [73]. In this case a mixture of ammonia water and ethanol was utilized as a solvent in the HA spinning solution. PLA12 utilized as a support medium was chosen because of the stability of the solution and repeatable fiber diameters (780 nm). Unfortunately, the authors did not demonstrate the difference between the PLA-scaffold with an HA-layer and without the one.

Thus, there are a several methods for obtaining electrospun nano- and microfibers based on hyaluronan: by the use of water-organic solvents; by the addition of surfactant, and/or by modifying polymers, by utilizing special techniques such as electroblowing; by using of the HA-derivatives; or by a combination of these methods. The independent fields are the utilization of specific additives such as kanamycin or metal nanoparticles, and post-treatment of the fibers obtained; for instance, cross-linking.

It is worth noting that the results obtained from research are difficult to compare each other because of different analysis methods utilized. Several studies [57,59,60,61,74] included only physical and rheological parameters of polymer solutions, and physicomechanical, physicochemical, and morphological characteristics of the fibers obtained. By contrast, some studies, for instance, [67,70], investigated the morphology, physicochemistry, and the biological effectiveness of each fiber. Other studies include a broad spectrum of analysis: from physicomechanical properties to antibacterial activity.

Given all, a brief knowledge overview from scientific sources is given in Table 1 (hyaluronic acid fibers without addition polymers) and Table 2 (fibers with modifying polymers).

## 3. Characterization

Firstly, initial polymer solutions have been analyzed by various methods to determinate the main characteristics which are demonstrated in Figure 3.

Note, that the solution spinnability and the fiber properties directly depend on the technological parameters of spinning process and the equipment used. In the case of wet spinning process, the spinnability is related to the spinneret diameter and the solution feed velocity, just as the fiber morphology depends also on the later processing (coagulation, stretching, drying, etc.). By contrast, electrospinnability and fiber properties are influenced by various technological parameters which are demonstrated in Figure 4. 

The fibers obtained come under all-around investigation, from surface morphology to biological activity measurements. Examples of the main parameters are shown in Figure 5.

The map of investigation is different from study to study, and comparative analysis of the results obtained is hindered. Table 3 illustrates the difference between the wet spinning studies reviewed. The majority of research projects are focused on the physicomechanical properties, morphologies, and biocompatibilities of the fibres obtained.

Fibre materials obtained by the electrospinning technique are characterized by a wider spectrum of measurement and analysis. The brief information from the studies is demonstrated in Table 4.

## 4. Main Biomedical Applications of HA Fibers and Particles

As mentioned above, the hyaluronan-based materials have a great variability, biocompatibility, and exploitability and could be successfully utilized as the key element in biomedical and bioengineering applications. There is not a large amount of scientific data related to biomedical applications of HA-fibers in comparison with, for example, chitosan, collagen or polycaprolactone-based ones. On the other hand, nanoparticles based on hyaluronan are well-studied and widely described as drug delivery systems for cancer treatment. It seems reasonable to take these results as a basis (“model”) for recognizing the bioactivity of hyaluronic acid-based materials in general.

### 4.1. Applications of HA-Based Nanoparticles

The huge number of investigations have designed nanoparticles with specific ligands (called active targeting ligands) which have improved the efficacy of drug delivery to specific tumor sites [82].

Hyaluronic acid is widely used polymer as a carrier for drug delivery, due to its biodegradable, biocompatible, non-inflammatory, and non-toxic properties. HA 0.48 MDa was used for the preparation of losartan-HA micelles [83]. Firstly, HA conjugated to 5β-cholanic acid was prepared. Such micelles of size 300 nm targeted hepatic stellate cells and demonstrated significant blockage of hepatic fibrosis in in vitro and in vivo studies.

HA/curcumin nanomicelles were applied in rheumatoid arthritis therapy [84]. The process for the preparation of micelles was carried out in dark to avoid the degradation of curcumin. HA (100 mg) was dissolved in DMF (30 mL), and DMSO (25 mL) was added at 60C; then, under N_2_ flow DCCI and DMAP were added. Finally, curcumin in DMSO was added. The resultant solution was frozen at −55 °C for 2 h, and then dried by sublimation in a vacuum chamber (1 Pa) for 24 h. The dry sample was stored in a refrigerator at 4 °C for further study. HA was used with molecular weights 10 and 25 kDa. Curcumin is bound to HA by ester linkages, and forms spherical nano micelles of 164 nm diameter, which show excellent biocompatibility, and thus, effectively promote the proliferation of chondrocytes.

In the recent reviews biomedical applications of HA-based nanomaterials for cancer treatment are well presented [85,86].

HA has been shown to not elicit any immune response in the body and can be used as an efficient delivery system for cancer treatment [87,88]. Micelles, polymersomes, and hydrogels have been utilized as HA-based nanocarriers and NIR-responsive materials, such as NIR dyes, gold nanoparticles, graphene oxide nanoparticles, Prussian blue nanoparticles, and magnetic nanoparticles, have been widely used for cancer treatment and water solubility [86,88].

IR–780-loaded HA-based micelles (HA–IR–780) were synthesized by a simple dialysis method for PTT (photothermal therapeutic) effects in TC-1 lung cancer cells [89]. HA with *M*_W_ 0.48 MDa was used, and the HA-IR 780 micelles with a size below 200 nm were obtained. The activity of micelles was confirmed on TC-1 xenografts in mice models. To investigate the CD44-specific cellular internalization of HA-IR 780 micelles, TC-1 cells were chosen due to their higher CD44 expression levels.

Self-assembled tumor-targeting HA–IR–780 nanoparticles (HA *M*_W_ = 10 kDa, and the average size was 171.3 ± 9.14 nm) for PTT in a CD44-over-expressed orthotopic bladder cancer model (MB-49 bladder cancer cells were used) have been developed [90]. Their activity was confirmed in vivo in C57BL/6 mice.

NIR-conjugated HA nanoparticles (NPs) (HA, *M*_W_ = 50 kDa, and the average diameter of the particles was 125 nm) have been synthesized for PTT. Superior anti-cancer effects and an improved life span, compared to the control group [91] were demonstrated both in vitro on A549 and NIH3T3 cells and in vivo on A549-bearing tumor mouse models. NIR dye-conjugated HA NPs (from HA with *M*_W_ 32kDa) and an encapsulation of perfluorooctylbromide NPs were synthesized for enhanced PTT. These nanoparticles had hydrodynamic diameters of 100 ± 10.7 nm, and selectively targeted CD44-over-expressing HT-29 human colon cancer cells both in in vitro and in vivo models [92].

Wang et al. [93] reported a rationally designed nanosystem based on a gold nanostar/siRNA against HSP72 of HA for selectively sensitizing MDA–MB–231 cancer cells. These nanosystems were synthesized by a surfactant-free method and HA with a molecular weight of 10 KDa was used. In comparison to the case without an HA coating, GNS/siHSP72/HA showed improved pharmacokinetics, with high accumulation in the tumor. These nanoparticles (particle size was 73.2 ± 3.8 nm) selectively target cancer cells by CD44 endocytosis and have an anti-cancer effect after laser irradiation in in vitro and in vivo models. Other groups have synthesized HA-modified Fe3O4@Au core/shell nanostars (298.8 and 339.4 nm) for tri-mode imaging and PTT in HeLa cancer cells [94]. HA with *M*_W_ 31,200 was used.

Biodegradable and reduction-sensitive nanocomplex was developed from arginine based poly(ester amide)s and HA (*M*_W_ = 10–20 kDa), and the PCI (photochemical internalization)-photosensitizer AlPcS2a was conjugated to the surface of the nanocomplex (ArgPEA-ss-HA(AP)). The average TEM diameter was detected as 57.2 ± 6.9 nm, and the average DLS (dynamic light scattering) diameter was detected as 109.9 ± 5.7 nm. This nanocomplex was used for the co-delivery of both PCI-photosensitizers and therapeutic agents to eliminate the biodistribution discrepancy resulting from the separated administration of free therapeutics. The PCI effect of the ArgPEA-ss-HA(AP) nanocomplex was validated in both monolayers and 3D spheroid models of MDA-MB-231 breast cancer cells. Synergism was detected between the PCI effect and doxorubicin-loaded nanocomplex in the inhibition of MDA-MB-231 cells. The ArgPEA-ss-HA(AP) nanocomplex provided enhanced intratumoral penetration in 3D spheroids compared to free AlPcS2a. The in vivo results suggested that the conjugation of AlPCs2a in the nanocomplex enabled the consistent and preferential accumulation of both doxorubicin and AlPcS2a in tumor sites. A light-enhanced anti-tumor effect was observed for the doxorubicin-loaded nanocomplex at well-tolerable dosage [95].

HA-conjugated AuNP-coated poly(glycidylmethacrylate) nanocomposites (PGMA@Au–HA) were fabricated for PTT with an average diameter of 200 nm [96], and tested on squamous cell carcinoma.

High or low molecular weight (380 and 102 kDa) HA-coated gold nanobipyramids (GBPs@h–HA and GBPs@l–HA, respectively) were synthesized for PTT in MDA–MB–231/Luc cells. GBPs@h–HA (the hydrodynamic diameter was 112.1 ± 1.0 nm) demonstrated superior targeting ability, when compared with GBPs@l–HA by HA–CD44 endocytosis. Upon 808 nm laser irradiation, a higher therapeutic efficacy was observed in the group GBPs@h–HA, compared to GBPs@l–HA, in both in vitro and in vivo studies [97]. The molecular weight of HA plays a key role in targeting and photothermal efficacy in CD44-over-expressing cancerous cells. It was shown that cellular uptake of GBPs@h-HA was 3.46 times higher compared to GBPs@l-HA. HA (380 kDa)-coated gold nanobipyramids possess more significant therapeutic efficacy in vivo.

One group [98] developed HA (*M*_W_ 9.27 kDa) functionalized-reduced graphene oxide (rGO) (an average lateral size was 108 ± 51 nm) for targeted cancer PTT. MCF-7 cells (CD44 overexpressing cell line) and NHDF (low CD44 expression) cells were used, also demonstrating a higher uptake by MCF-7 cells. A nanographene oxide (NGO)–HA (*M*_W_ 100 kDa) conjugate (NGO–HA) was developed with a lateral size of ca. 200 nm and a thickness of ca. 1.04 nm. It was applied for PTT of melanoma skin cancer [99].

Chondroitin sulfate (CS)-HA/IgA nanoparticles have been obtained and the average size was between 425–665 nm with a positive zeta potential. These results pointed out that CS-HA can be an effective carrier for use in targeted drug delivery [41]. HA’s molecular weight was 1500 × 10^3^ g/mol. Low molar mass HAs (580 × 10^3^ g/mol, 120 × 10^3^ g/mol, 50 × 10^3^ g/mol were obtained by sonication.

mRNA was successfully delivered with chitosan/HA nanoparticles (HA *M*_W_ 680 KDa) for the first time for the treatment of human colorectal carcinoma cell line HCT-116 [43].

Novel wound dressings composed of non-woven cotton fabric and multilayer of HA (*M*_W_ 1.48 × 10^6^ Da) and chitosan were built using layer-by-layer assembly technique. Moreover, to enhance the antibacterial properties of the aforementioned dressings, silver nanoparticles (Ag NPs) were prepared and incorporated as a functional additive in the final HA layer of such dressings. The TEM image of the prepared Ag-NPs depicts that the particle size of that nano-particles was <13 nm [100].

New, broadly adaptable materials platforms are based on HA (20 kDa and 120 kDa) for mimicking key extracellular matrix features of viscoelasticity and fibrillarity in hydrogels for 3D cell culture and shedding light on how these mechanical and structural cues regulate cell behavior [101]. By increasing HA molecular weight, the hydrogel modulus was enhanced while the relaxation rate was reduced.

### 4.2. Biomedical Application of HA-Based Fibers

Electrospun nanofibrous scaffolds from gelatin and chemically sulfated or non-sulfated hyaluronan (sHA or HA) (HA *M*_W_ 1.1 × 10^6^ g/mol) and chondroitin sulfate (CS) containing chemically sulfated glycosaminoglycans (GAGs) at different concentrations have been developed. Evenly distributed fiber morphology was observed with no differences between varying concentrations and types of GAGs. An average diameter of fibers was 260 ± 32 nm. In vitro release kinetics revealed that GAGs release is driven by diffusion. The effects of these scaffolds were analyzed on human keratinocyte, fibroblast (Hs27) and mesenchymal stem cell (hMSCs) adhesion and proliferation. A significant increase in cell number (~5 fold) was observed when cultivating all three cell types alone on scaffolds containing sHA and CS [102]. The discussed above study is summarized in Figure 6.

HA/poly(lactic-co-glycolic acid, PLGA) core/shell fiber matrices loaded with epigallocatechin-3-O-gallate (EGCG) (HA/PLGA-E) are fabricated by coaxial electrospinning (HA *M*_W_ 0.8–1.8 × 10^6^ Da). HA/PLGA-E core/shell fiber matrices are composed of randomly-oriented sub-micrometer fibers and have a 3D porous network structure. The average diameters of HA/PLGA, HA/PLGA-E, and HA/PLGA-E fibers were 519 ± 88, 365 ± 152, and 292 ± 147 nm, respectively. EGCG is uniformly dispersed in the shell and sustainably released from the matrices in a stepwise manner by controlled diffusion and PLGA degradation over four weeks. EGCG does not adversely affect the thermomechanical properties of HA/PLGA-E matrices. The number of human dermal fibroblasts attached on HA/PLGA-E matrices is appreciably higher than that on HA/PLGA counterparts, while their proliferation is steadily retained on HA/PLGA-E matrices. The wound healing activity of HA/PLGA-E matrices was evaluated in streptozotocin-induced diabetic rats. After two weeks of surgical treatment, the wound areas were significantly reduced by the coverage with HA/PLGA-E matrices resulting from enhanced re-epithelialization/neovascularization and increased collagen deposition, compared with no treatment or HA/PLGA. The wound healing activity of the HA/PLGA-E matrices can be attributed to the synergistic effects of HA and EGCG, which prominently promote the re-epithelialization, ECM re-organization, and revascularization in wounds [103].

In order to reduce the viscosity of HA, two different methods were used: addition of sodium chloride and blending with biocompatible polymers that readily electrospin, such as PEO and PVA [104]. HAs with molecular weights of 1.5 million and 1 million, and low molecular weight HA *M*_W_ = 680,000 were used. Both 1.0 wt% to 2.0 wt% HA solutions have appeared to be optimal concentration ranges for electro-spinning of HA. The electrospinning results from the lower molecular weight (*M*_W_ = 680,000) HA samples do not produce nanofibers as the polymer chains are not sufficiently entangled to form a fibrous morphology. No fiber formation was observed when 1M NaCl was added to 1.5% HA solution and in the case 2 wt% HA with 0.5 M NaCl. The blended HA-PEO solutions (60/40 and 70/30) produce fibers with beaded structures.

A bilayered, nonwoven material for tissue regeneration was prepared from chitosan (CS) and HA (*M*_W_ 5.4 × 10^4^) by needleless electrospinning, wherein 10–15 wt% (with respect to polysaccharide) polyethylene oxide was added as spinning starter [76]. The roughness of the bilayer material was in the range of 1.5–3 μm, which is favorable for cell growth. Electrospinning resulted in the higher orientation of the polymer structure compared with that of corresponding films, and this finding may be related to the orientation of the polymer chains during the spinning process. These structural changes increased the intermolecular interactions. The bilayer CS-HA scaffold exhibited good compatibility with mesenchymal stem cells. This characteristic makes the material promising for tissue engineering applications.

Fabricated hybrid microfibers composed of HA (*M*_W_ 41–65 kDa) and multiwalled carbon nanotubes (MWCNT) by a wet-spinning method have been prepared [45]. HA acts as a biosurfactant and an ionic crosslinker, which improves the dispersion of MWCNTs and helps MWCNT to assemble into microfibers. The HA/MWCNT hybrid microfibers with excellent electrical conductivity, mechanical properties, and stable behavior are a promising material for microsized materials, conductive materials, electrode materials, intelligent materials, and high-performance, fiber reinforced composite materials.

The obtained crosslinked HA/SWCNT (single-wall carbon nanotubes) hybrid microfibers (an approximate diameter of 50 μm at a dry state) show an excellent capacitance and actuation behavior under mechanical loading with a low potential of ±1 V in a biological environment. Furthermore, the HA/SWCNT microfibers exhibit an excellent in vitro viability. Finally, the biocompatibility was shown through the resolution of an early inflammatory response less than 3 weeks after the implantation of the microfibers in the subcutaneous tissues of mice [105].

Peptides with RGD, IKVAV, or SIKVAV adhesive motif were attached to HA-based fiber (HA *M*_W_ 0.83 MDa) or non-woven textile through ester bond using solid phase peptide synthesis. A linker between HA and peptide containing three glycine or two 6-aminohexanoyl units was applied to make peptides more available for cell surface receptors. Dermal fibroblasts adhered readily on this material, preferentially to RGD peptide with 6-aminohexanoyl linker. Contrarily, the absence of adhesive peptide did not allow cell attachment but maintained the material stability [106]. The outlines of this research are demonstrated in Figure 7.

During the preclinical study of HA nanofibers on pig wounds’ healing when covered by adhesive bandages, a sterilized solid HA, gauze with Vaseline, an antibiotic dressing, and a sterilized HA nanofiber wound dressing were compared [58]. Cocamidopropyl betaine was added to the HA solution as a surfactant.

Silver (Ag) nanoparticles embedded in electrospun HA/polycaprolactone (PCL) nanofibrous membranes (NFMs) (HA/PCL+Ag NFMs) to prevent peritendinous adhesions and bacterial infection after tendon surgery were developed. HA (1.3 × 10^6^ Da) was used for effective lubrication, and Ag provided antibacterial activity. Polycaprolactone NFMs, HA/PCL core-sheath NFMs, and HA/PCL + Ag NFMs with comparable fiber diameters (344 ± 92 nm) and pore sizes (1.40 ± 0.84 nm) were prepared and analyzed [107].

Hyaluronan fiber was prepared by a wet-spinning technique (HA MW 1.75 MDa). Consequently, hyaluronan fiber was used as a capping and stabilizing agent for the preparation of fibers with silver nanoparticles. HA-Ag NPs showed high antibacterial activity against *Staphylococcus aureus* and *Escherichia coli*. Cell viability tests indicated that hyaluronan, hyaluronan fibers, and hyaluronan fibers with silver nanoparticles (Ag NPs 10 and 40 nm) were non-toxic on the cell growth using mouse fibroblast cell line NIH 3T3 [108].

HA (MW = 0.8 − 1.8 × 10^6^ Da)/poly(lactic-co-glycolic acid, PLGA) core/shell fiber meshes loaded with epigallocatechin-3-O-gallate (EGCG) (HA/PLGA-E) were prepared via coaxial electrospinning. The diameter of HA/PLGA fibers was found to range between 790 nm and 1578 nm (average fiber diameter: 1270 ± 510 nm). After 28 days, the amount of EGCG released from HA/PLGAE meshes achieved a constant rate, reaching about 70% [109].

Fabrication of a mechanically stable, biocompatible, bilayered polymeric scaffold consisting of chitosan (CS)/polycaprolactone (PCL) and HA (MW 1000–2000 kDa) using a less toxic solvent system (DMF) was done. The electrospinning technique was used followed by morphological, physiochemical, and mechanical characterizations. The average fiber diameter of CS/PCL-HA bilayered scaffold was found to be 362.2 ± 236 nm which is in the range of collagen fiber found in the ECM. Enhanced swelling, degradation, hydrophilicity, and water vapor transmission rate were found for the bilayered scaffold compared to that of the PCL and CS-PCL scaffolds. Antimicrobial property evaluation revealed reduction in bacterial adhesion on bilayered scaffolds. The growth inhibition of *E. coli* was observed. In vitro studies with vero cells (kidney epithelial cells, extracted from African Green Monkey (*Chlorocebus* sp.)) confirm enhanced proliferation, growth, and migration of vero cells on the bilayered CS/PCL-HA scaffold to that of PCL and CS/PCL scaffolds [73].

The polycaprolactone (PCL) spiral structure was surface functionalized with PCL nanofibers encapsulated with chondroitin sulfate (CS) (20% (*w*/*w*)) and HA (0.2% (*w*/*w*)). In order to retain and sustain the release of CS and HA, nanofibers were cross-linked using carbodiimide chemistry [110]. The average fiber diameter was 100–130 nm. Such fibers may be used in cartilage tissue engineering.

A HA fiber scaffold with an extracellular-matrix-mimetic structure has been prepared via lyophilization. To enhance the water-resistance, the pure HA fiber membranes were chemically cross-linked by 1-ethyl-3-(3-dimethylaminopropyl) carbodiimide hydrochloride (EDC) [111]. HA with MW = 2,000,000 was used. The fiber membranes were prepared by freezing HA solutions at different concentrations 0.05 wt%, 0.15 wt%, and 0.3 wt%, and then cross-linking with EDC for 24 h. The formation of a sheet-like structure might be caused by the collique-faction of the HA fibers on the surfaces of ice crystals due to the high concentrations. The water resistant property of the crosslinked fibers was enhanced as the increase of the cross-linking time. The water resistant property of the fibers after 24 h cross-linking was increased, such that about 89% remained after 48 h degradation in the water, and 71% at 72 h; but unfortunately, almost 100% of the 24 h cross-linked fibers were degraded by immersion in the PBS solutions for 48 h.

The bending Young’s modulus was determined using atomic force microscope by involving single-material (PVA, (PEO 400K)) and composite nanofibers (PVA/HA), EO 400K/CS)) [112]. HA with MW 400,000 g/mol was used. The mechanical property, namely, the bending E, increases as the diameters of the fibers decrease from the bulk down to the nanometer regime (less than 200 nm). PVA, PVA/HA, and PEO nanofibers are more elastic (a smaller bending Young’s modulus), and therefore are the most suitable for skin and wound tissue scaffolds. The design trend in tissue-engineering scaffolds is presented in Figure 8.

The mechanics (through intrafiber crosslink density) and adhesivity of electrospun HA fibers significantly affect human mesenchymal stem cell (hMSC) interactions and gene expression [113]. A 1% *w*/*v* solution of HA (64 kDa) was reacted on ice with either 0.67 mL (for 35% modified) or 2.23 mL (for 100% modified) methacrylic anhydride with maintenance of pH at ~7.5–9 for 1.5 days. Cysteine-containing RGD peptides (GCGYGRGDSPG) were conjugated to MeHA via Michael addition between thiols on the peptides and methacrylates on MeHA. MeHA solutions for electrospinning were composed of 4% *w*/*v* MeHA, 2% *w*/*v* poly(ethylene oxide) (PEO, Sigma, 900 kDa), and 0.5% *w*/*v* Irgacure 2959 in deionized water. Fiber diameters of dry fibers were 186 ± 7 nm for 35% modified MeHA and 177 ± 6 nm for 100% modified MeHA, with a notable increase in fiber diameter upon swelling to 601 ± 36 and 744 ± 45 nm for 100% and 35% modified.

In most publications it is indicated that hyaluronic acid can target tumor cells through CD44 receptor-mediated endocytosis, and this mechanism was confirmed by several studies. In the case of HA nanoparticles, authors mainly used low-weight hyaluronan, while high weight HA is used in the preparation of HA based nanofibers. Particles based on hyaluronic acid with a molecular weight of from 10 KDa before 50 KDa can be successfully used in tumor therapy due to high activity and interaction with receptors on the tumor cells’ membranes.

Fibers, in turn, on the basis of hyaluronic acid, have regenerative, sorbing, and antibacterial properties, and therefore have prospects when used as wound coatings, surgical dressings, etc.

## 5. Conclusions

Biopolymer hyaluronic acid has attracted the attention of chemists, bioengineers, physicians, and other experts and scientists from the date of its discovery. Some of the most challenging and topical ways of utilizing hyaluronic acid are in nano- and microfibers, micelles, and nanoparticles. Materials based on such structures have wide applications, from wound healing to scaffolds for drug delivery systems. This review is a brief summary of the latest discoveries and developments, which could be helpfully for the elaboration of novel and modified hyaluronic-acid fibrous materials with unique properties.

## Figures and Tables

**Figure 1 polymers-11-02036-f001:**
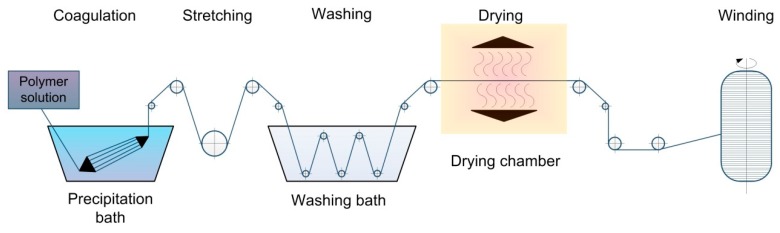
The schematic view of the wet spinning process.

**Figure 2 polymers-11-02036-f002:**
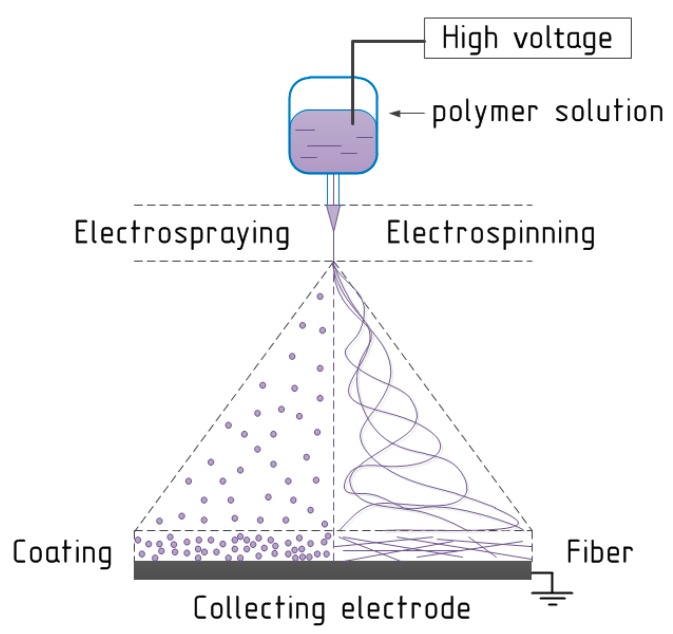
The schematic view of the electrospinning/electrospraying process.

**Figure 3 polymers-11-02036-f003:**
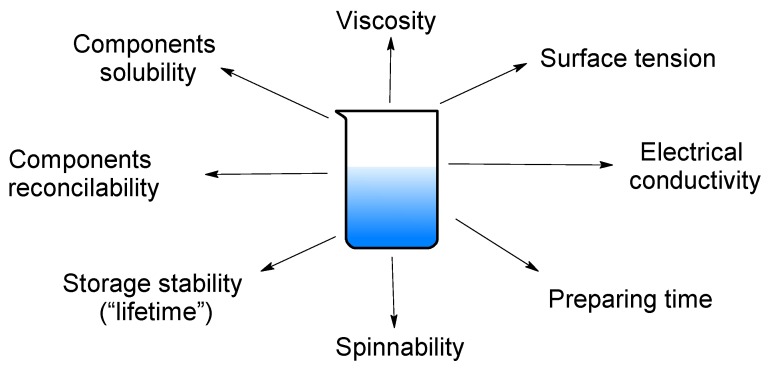
The key detectable parameters of polymeric solutions.

**Figure 4 polymers-11-02036-f004:**
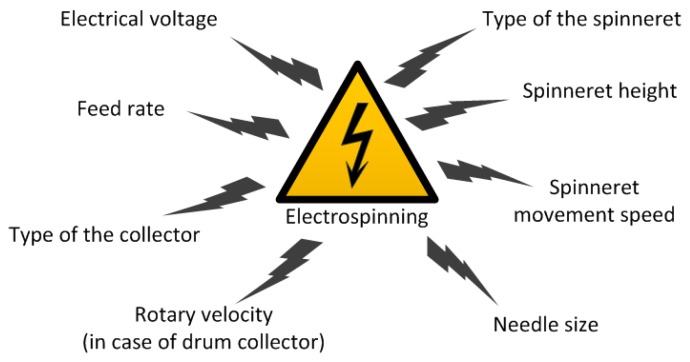
The key technological parameters of the electrospinning process.

**Figure 5 polymers-11-02036-f005:**
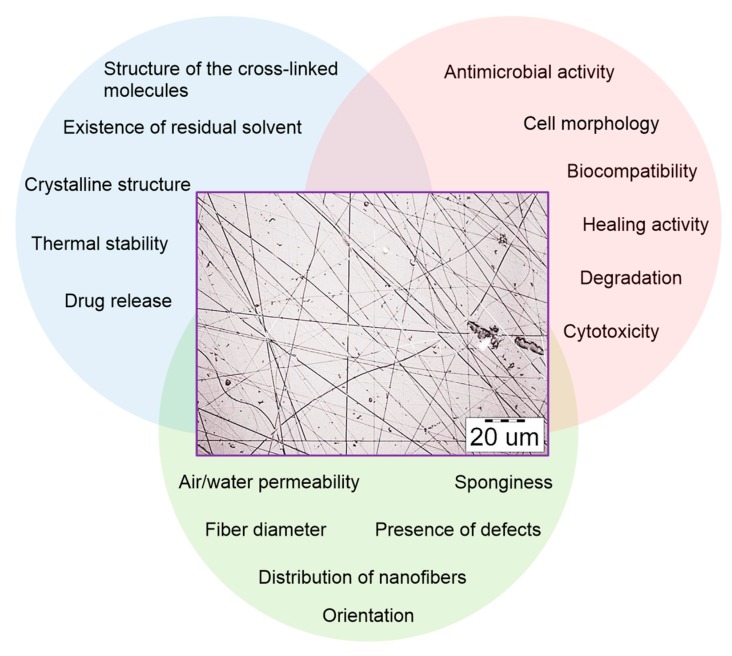
The key detectable parameters of polymer fibers.

**Figure 6 polymers-11-02036-f006:**
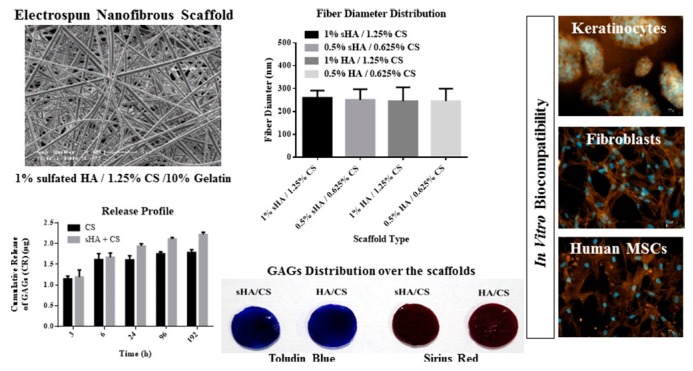
Biomimetic electrospun scaffolds from main extracellular matrix components for skin tissue engineering applications. Reprinted with permission from [102].

**Figure 7 polymers-11-02036-f007:**
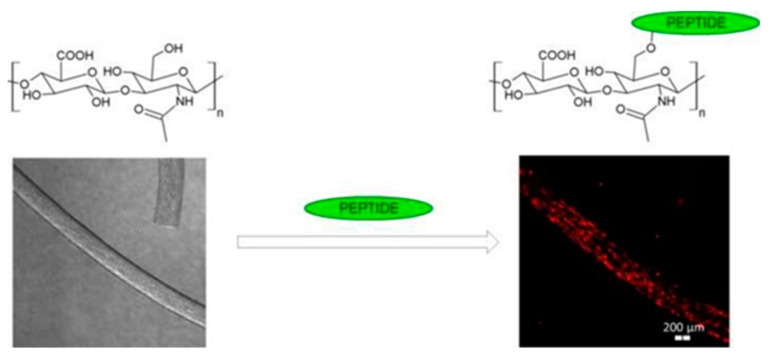
Cell-seeded hyaluronan-based scaffold with peptides for tissue engineering. Reprinted with permission from [106].

**Figure 8 polymers-11-02036-f008:**
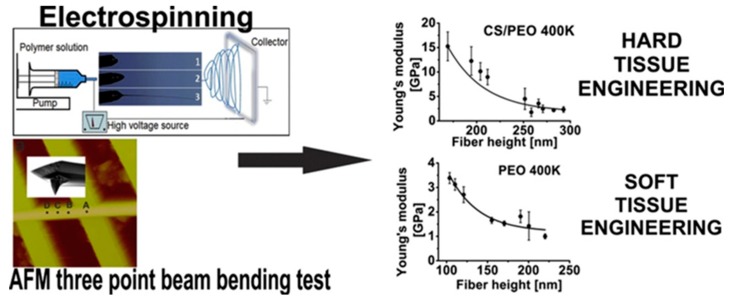
The design trend in tissue-engineering scaffolds based on nanomechanical properties of individual electrospun nanofibers. Reprinted with permission from [112].

**Table 1 polymers-11-02036-t001:** A variety of electrospinning solutions based on hyaluronan without carrying polymers.

*M*_w_ (HA), kDa	C_HA_, *w*/*v*%	Solvents (*v*/*v* or *w*/*w*)	Additives	References
2000	1.3–1.5	DW/ethanol (9/1) DMF/DW (2, 1.5, 1, 0.5) DMF/DW (1.5)	- - GE (80 kDa)	[57]
n.r.	1.0–4.0	DW	Cocamidopropyl betaine	[58]
1000	0.8–1.2	DW/FA/DMF (25/50/25)	-	[59]
2.6–2.7	0.75	DMF/DW (0:1, 0.25:1, 0.5:1, 1:1)	-	[60]
2000	3.0 1.5	0.5 M NaOH/DMF (4:1) NH_4_OH/DMF (2:1)	-	[61]
3500 45	0.01–3.0 1.0–2.0	HCl (pH = 1.5)	Ethanol	[18,19]
1300	1.5–2.1	DMSO/DW (from 0:1 to 3:2)	-	[63]

Abbreviated terms: DMF—dimethyl formamide; DMSO—dimethyl sulfoxide; DW—distillated (deionized) water; FA—formic acid; GE—gelatin (as surfactant); HA—hyaluronan/hyaluronic acid; n.r.—not reported.

**Table 2 polymers-11-02036-t002:** A variety of electrospinning solutions based on hyaluronan with additional polymers.

*M*_w_(HA), kDa	AdP (M_w_), kDa	C_HA_, *w*/*v*%	C_AdP_, *w*/*v*%	Solvents (*v*/*v* or *w*/*w*)	Additives	References
8.7	PEO (900)	Σ 8.0		AA/DW (50:50)	-	[64]
600–1100	PEO (200)	0.1–0.3	12.0	DW	Kanamycin (1.0 % *w*/*v*)	[65]
680	CL (*M*w n.r.)	Σ 10.0 (WR 95/5, 80/20)		NaOH/DMF (4/1)	NaCl (salt particulates) EDC	[66]
HA-DTPH (158)	PEO (900)	2.5	2.5	DMEM	PEGDA	[70]
57	PVA (130)	Σ 6.0		DW	HPβCD EDC, NHS	[72]
601–850	CL (*M*w n.r.)	Σ 7.5 (WR 95/5)		NaOH/DMF (4/1)	Au nanoparticles EDC	[67]
1000– 2000	PEO (900)	0.2	5.0	DW	-	[73]
1000	CS (200)	1.0	7.0	DW/FA (25/75) DW/FA (20/80)	-	[74]
54.0	PEO (*M*w n.r.)	4.0	0.6	DW	-	[76]
HA-DTPH (158)	PEO (900)	2.0	0.5–2.0	DMEM	PEGDA	[78]
Nor-HA (*M*w n.r.)	PEO (*M*w n.r.)	3.25	2.75		Albumin, UV-initiator, DTT	[79]

Abbreviated terms: AA—acetic acid; AdP—additional polymer; CL—collagen; CS—chitosan; DMEM—Dulbecco’s Modified Eagle’s Medium; DMF—dimethyl formamide; DMSO—dimethyl sulfoxide; DTT—dithiothreitol; DW—distillated (deionized) water; EDC—1-ethyl-3-(3-dimethylaminopropyl)carbodiimide hydrochloride; FA—formic acid; GE—gelatin; HA—hyaluronan/hyaluronic acid; HA-DTPH—3,3′-dithiobis(propanoic dihydrazide)-modified hyaluronic acid; HPβCD—(2-Hydroxypropyl)-β-cyclodextrin; n.r.—not reported; NHS—N-Hydroxysuccinimide; Nor-HA—norbornene-functionalized hyaluronic acid; PEGDA—Poly(ethylene glycol) diacrylate; PEO—polyethylene oxide; PVA—polyvinyl alcohol; WR—weight ratio.

**Table 3 polymers-11-02036-t003:** The parameters measured and the methods utilized (for the wet spinning technique).

Properties/Methods	References							
	[33]	[34]	[35]	[36]	[37]	[38]	[39]	[45]
Electrical conductivity of fibers								+
Fiber processability		+						
Physicomechanical properties		+	+		+	+	+	+
Morphology/microstructure		+	+		+	+	+	+
Porosity							+	
Thickness	+				+	+		
Fineness		+	+					
Surface density of functional groups				+				
Tissue regeneration (*in vivo*)							+	
Immunohistochemical analysis					+	+	+	
Ion-exchange high-performance liquid chromatography			+					
Gas-chromatography with flame ionization detector			+					
Infrared spectroscopy (FTIR)								+
UV-Vis spectrophotometry				+				+
X-ray diffractometry	+							
Differential scanning colorimetry								+
Thermo gravimetric analysis								+
Antimicrobial activity				+				
Degradation (*in vitro*)			+					
Biocompatibility/cytocompatibility			+	+	+	+	+	
Swelling		+	+					
Flexibility								+
Stability								+

**Table 4 polymers-11-02036-t004:** The parameters measured and the methods utilized (for the electrospinning technique).

Properties/Methods	References																			
	[18]	[19]	[57]	[58]	[59]	[60]	[61]	[63]	[64]	[65]	[66]	[67]	[70]	[72]	[73]	[74]	[76]	[78]	[79]	[81]
Solution viscosity	+	+	+		+			+	+					+		+				+
Surface tension of solution			+		+	+			+					+						+
Electrical conductivity of solution			+		+	+	+		+					+		+				
Spinnability/stream stability	+	+	+	+	+	+	+	+		+	+	+		+	+	+	+			
Physicomechanical properties											+			+	+		+			
Morphology/microstructure	+	+	+	+	+	+	+	+	+	+	+	+	+	+	+	+	+	+	+	+
Porosity											+				+		+			
Degree of crystallinity				+					+											
Thickness				+																
Air/liquid permeability				+											+					
Wound healing (*in vivo*)				+																
Infrared spectroscopy (FTIR)		+				+	+		+	+		+	+	+	+			+		+
UV-Vis spectrophotometry														+						
X-ray photoelectron spectroscopy							+		+											
X-ray diffractometry									+								+			
Atomic force microscopy																	+			
Differential scanning colorimetry									+	+			+			+		+		
Thermo gravimetric analysis									+	+								+		
Cytotoxicity									+					+						
Antimicrobial activity										+					+					
Degradation (*in vitro*)											+				+					
Biocompatibility/cytocompatibility											+	+	+	+	+		+	+	+	+
Drug concentration/drug release										+				+						
Swelling		+											+	+	+		+	+	+	
Buffer ability															+					
Wettability															+					+
Free surface energy																				+

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
