# Peer review of "Hyaluronan-Based Nanofibers: Fabrication, Characterization and Application"

_polymers, 2019, doi:10.3390/polym11122036_

Round 1

Reviewer 1 Report

I thank authors for correcting and improving the quality significantly. I recommend to publish the review as is.

Author Response

We are thankful to Reviewer 1 for the favourable decision and the manuscript approval.

Reviewer 2 Report

I have read carefully the review and the changes that have been done after reviewer comments. The paper has been substantially enhanced and summarizes the recent applications of hyaluronic acid nano- and microfibers preparation and their applications. I think that the paper is appropriate for publication. I propose to authors to add some figures (2-3) from literature conserving the application of these nanofibers.   

Author Response

We are thankful to Reviewer 2 for the favourable decision.

We added three Figures (6-8).

This manuscript is a resubmission of an earlier submission. The following is a list of the peer review reports and author responses from that submission.

Round 1

Reviewer 1 Report

The current review is focused on hyaluronic acid based fiber fabrication Line 10: Remove word "modern"
Line 61: Correct sentence to "Despite the wide usage of"
Lines 62, 63: Correct the sentence grammar
Line 65: Some of the latters?
Line 66: considered in "detail" further
Line 67: fabrication "of"
Line 82: "Melt" spinning process
Authors are seriously recommend to consider help from a native english speaker to rewrite the whole review article. Draft is difficult to follow due to all types of grammar issues through out the document. Figure 2 is unnecessary as the electrospinning and electrospraying techniques are widely reviewed in the past. Section 2 has little emphasis on HA fiber fabrication and fiber properties. Much of the text speaks about HA solvent systems, chemical modification of HA, the general polymer fiber techniques which are widely reviewed in the past in many publications. Very little discussed on fiber characterization (just two lines and a figure) Section 3 is missing Section 4.1 speaks about HA nanoparticle applications and is its relevance to the goals of review is not clear.
Section 4.2 discussed about HA fiber applications but largely failed to summarize the progress.
Conclusion part should be elaborated. Current trends and future research requirements should be proposed to justify the purpose of a review article.     In the current form, this review is not suitable for publication

Reviewer 2 Report

This review article aim to review various methods for fabricating hyaluronic acid-based fibers and explore its physical, chemical and biological properties. Overall, this article is knowledgeable and helpful for a reference for the research who works on this field. However, the reviewer has the following comments which may helpful for improve this article.

On line 57, the author should note an error message on reference. As the author mentioned that high viscosity of hyaluronic acid solutions is a shortcoming and be a problem need to overcome for fabricating hyaluronic acid-based fibers. It is well known that molecular weight strongly affect the physical property of hyaluronic acid, including viscosity. Although in the Table 1 and 2, the author listed the MW that the reviewed literatures used, a short discussion on the MW issue should be provided in the text The application of Low-molecular weight hyaluronic acid was discussed by several scholars. I suggest the author added a paragraph to discuss this material.

Reviewer 3 Report

This paper is unsuitable for publication. The overall organization and presentation is poor. Most figures do not add to understanding of the concepts. Figure three and four should be removed. The use of single-sentence paragraphs is confusing. The conclusions section is weak and does not contain any useful conclusions.

Reviewer 4 Report

COMMENTS AND SUGGESTIONS ARE INCLUDED IN THE ATTACHED FILE
